# Spontaneous current constriction in threshold switching devices

Jonathan M. Goodwill [1,2], Georg Ramer[2], Dasheng Li[1,2], Brian D. Hoskins[2], Georges Pavlidis [2], Jabez J. McClelland [2], Andrea Centrone [2], James A. Bain[3] & Marek Skowronski[1]

Threshold switching devices are of increasing importance for a number of applications including solid-state memories and neuromorphic circuits. Their non-linear characteristics are thought to be associated with a spontaneous (occurring without an apparent external stimulus) current flow constriction but the extent and the underlying mechanism are a subject of debate. Here we use Scanning Joule Expansion Microscopy to demonstrate that, in functional layers with thermally activated electrical conductivity, the current spontaneously and gradually constricts when a device is biased into the negative differential resistance region. We also show that the S-type negative differential resistance $I$–$V$ characteristics are only a subset of possible solutions and it is possible to have multiple current density distributions corresponding to the same value of the device voltage. In materials with steep dependence of current on temperature the current constriction can occur in nanoscale devices, making this effect relevant for computing applications.

[1] Department of Materials Science and Engineering, Carnegie Mellon University, 5000 Forbes Ave, Pittsburgh, PA 15213, USA. [2] Nanoscale Device Characterization Division, Physical Measurements Laboratory, National Institute of Standards and Technology, Gaithersburg, MD 20899, USA. [3] Department of Electrical and Computer Engineering, Carnegie Mellon University, 5000 Forbes Ave, Pittsburgh, PA 15213, USA. Correspondence and requests for materials should be addressed to M.S. (email: mareks@cmu.edu)

Recent commercialization of oxide- and chalcogenide-based solid-state memories has renewed interest in devices exhibiting volatile switching phenomena. Such devices serve a vital role as selection devices suppressing the sneak path currents in memory arrays[1], and are the backbone of ultra-compact oscillators in oscillatory neural networks[2]. Among many possible switching devices, of most interest are metal/oxide/metal (M/O/M) structures that display S-type negative differential resistance (S-NDR) I–V characteristics. Such I–V's have single valued voltage (one value of voltage for any value of current) but multiple values of current for voltage range between threshold ($V_{TH}$) and the holding voltage[3]. S-NDR, in turn, is frequently associated with the formation of high current density domains parallel to the electric field. The domains are commonly referred to as current filaments; here we adopt the nomenclature of current density domains and/or current constriction to differentiate this effect from permanent filaments found in memory devices. Recent work on S-NDR devices has been focused on oxides with thermally activated (Poole-Frenkel-type) electrical conductivity[4–10].

Long disputed questions in S-NDR type devices concern the conditions under which the uniform current would constrict into domains and the size of such constriction. Answering those questions became more important recently as the current constriction is thought to be the first step in the electro-formation process in resistive switching devices, which critically depends on temperature and temperature gradient distributions[10–14]. Early attempts to identify rules of domain stability such as minimum entropy production have not succeeded in establishing clear criteria[3]. In fact, Landauer argued it is impossible to define the stability based on the parameters of the steady states alone[15]. In a recent report, Kumar and Williams revisited the issue and proposed that the stability is determined by the minimum internal energy of the device[10]. These authors also argued that the domains could only form as a transient state during the threshold switching event when the differential circuit resistance decreases to zero and the device transitions between two points of the I–V at a constant source voltage ($V_{SOURCE}$). If the circuit is stabilized by a large load resistor, the current distribution was suggested to remain uniform for all values of voltage. This interpretation contradicted finite element modeling results that found current constriction to occur in the steady state[7]. However, an experimental confirmation of the current constriction in the S-NDR class of devices has not yet been provided.

Oxides that undergo insulator-to-metal transition (IMT), such as $VO_2$[16] and $NbO_2$ at high temperatures[9,10], also exhibit highly nonlinear I–V's but of a different type, which we refer to as multivalued. These I–V's have two branches that are not connected with each other. The characteristics have multiple values of current corresponding to one value of voltage, similar to S-NDR, but they also have multiple values of voltage for one value of current. Kumar and Williams proposed that, in addition to the current constriction, high electric field domains form in $VO_2$, similarly to N-NDR systems[3]. They also asserted that multivalued I–V's must be associated with an abrupt change of material properties during the phase transition[10]. So far, the two types of characteristics (S-NDR and multivalued) seem to be unrelated and quite distinct, appearing in systems with unmistakably different material properties.

In this paper, we present experimental evidence of the current constriction in the steady state in $TaO_x$-based M/O/M structures with S-NDR characteristics, in quantitative agreement with the results of an electro-thermal finite element model. Also, we demonstrate that multivalued I–V's can be obtained in devices characterized by thermally activated electrical conductivity with all other material properties being temperature independent. Notably, both S-NDR and multivalued I–V's are associated with the appearance of high current density domains and do not require formation of electric field domains.

## Results

**Experimental and simulated S-NDR I–V characteristics.** Figure 1a shows the quasi-DC I–V characteristics of a $TiN/TaO_x/TiN$ via-type device with the lateral size of 2 μm × 2μm and a functional layer thickness of 50 nm measured in a circuit that included a 107 kΩ ± 1 kΩ load resistor. Throughout the manuscript, the resistance uncertainty represents a single standard deviation in fitting the load resistor I–V slope. A quasi-DC voltage sweep is defined such that the time delay between the change of the voltage and the current measurement is much longer than the thermal and electrical time constants of the device. In all figures in this report, the current is plotted as a function of the voltage drop across the device ($V_{DEVICE}$) calculated as $V_{SOURCE} - I \times R_{LOAD}$, rather than $V_{SOURCE}$; $R_{LOAD}$ represents the load resistance (Supplementary Fig. 1).

Amorphous oxide-based devices typically display exponential dependence of current on applied voltage and thermally activated dependence on temperature. This behavior is frequently fitted with Poole-Frenkel model of conductivity[17,18]. This corresponds to linear I–V at low bias, becoming visibly super linear at voltages exceeding 2.5 V due to Joule heating and the field-induced reduction of effective activation energy of traps[7,19]. There is an obvious positive feedback between the current, conductance, and temperature of the device. With further increase in device current, the gain in the feedback loop between the current and the temperature increases, $\partial I/\partial V$ eventually diverges to infinity, and the I–V enters into the NDR region. Due to the large series load resistor, the total differential circuit resistance was always positive, the current was well defined, and changed gradually with the increase of $V_{SOURCE}$. The I–V curve displays no noticeable changes up to $10^9$ cycles if the current is limited to less than 350 μA and shows symmetry with respect to the bias polarity.

The blue dashed line in Fig. 1a shows the finite element simulation results. The simulation solved the 3D equations for the heat and charge flow, with the added restriction of the load resistor in the circuit[7]:

$$\rho C_p \frac{\partial T(x,y,z,t)}{\partial t} - \kappa \Delta T(x,y,z,t) = \frac{j^2(x,y,z,t)}{\sigma(x,y,z,t)} \tag{1a}$$

$$\nabla \cdot j(x,y,z,t) = 0 \tag{1b}$$

$$V_{SOURCE}(t) = I(t) R_{LOAD} + V_{DEVICE}(t) \tag{1c}$$

where $C_p$ is the heat capacity, $T$ the temperature, $j$ the current density, $\sigma$ the electrical conductivity, $\rho$ the mass density, $\kappa$ the thermal conductivity, and $I$ is the total current. Current density, temperature, and electrical conductivity are all functions of position and time while $V_{SOURCE}$, $I$, and $V_{DEVICE}$ depend only on time. The input parameters included the experimentally determined low field electrical conductivity and standard values of the material constants found in the literature. All essential details of the procedure are described in the Methods and in our previous publications[7,8]. The steady state solution for the $I(V_{DEVICE})$ reasonably well reproduces the shape of the I–V, especially considering that the simulation did not include any adjustable parameters.

Figure 1b shows an enlarged segment of the I–V obtained using a smaller load resistance of 3.9 kΩ ± 0.1 kΩ. In this case, the slope of the load line (given by $-1/R_{LOAD}$) is higher than the lowest slope of the intrinsic device characteristics within the NDR region (shown as a black trace, identical to that in Fig. 1a). As a result,

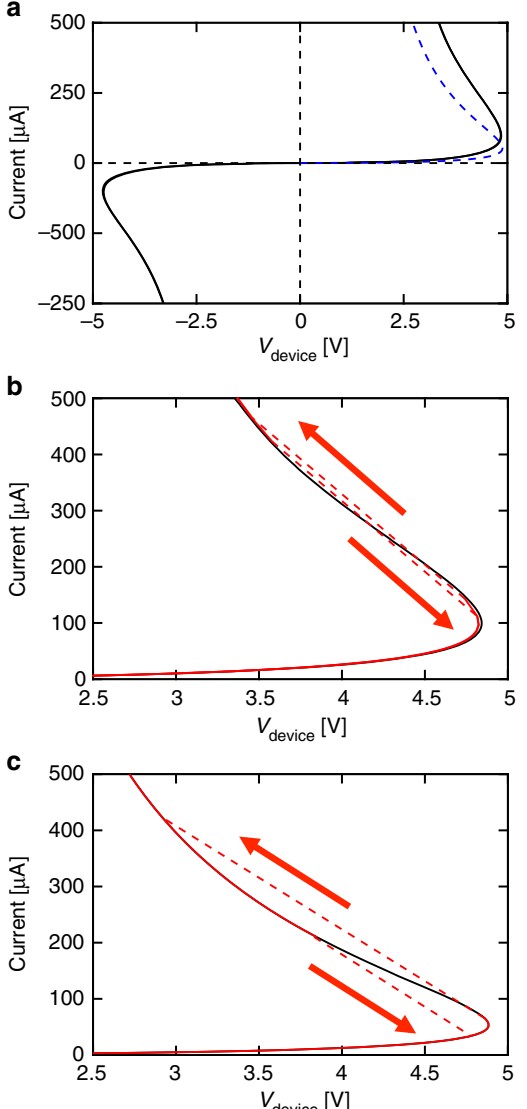

**Fig. 1** Simulated and experimental S-NDR in TaO$_x$ devices. **a** Quasi-DC experimental (continuous black line) and simulated (dashed blue line) *I–V* characteristics of the 2 μm devices with 107 kΩ ± 1 kΩ load resistor in series. **b** Magnified view of two experimental *I–V*'s with different values of load resistance. The black line is the *I–V* from (**a**) and the red line is the *I–V* obtained with $R_{LOAD}$ = 3.9 kΩ ± 0.1 kΩ. The smaller $R_{LOAD}$ induces threshold switching (red-dashed lines) with hysteretic behavior of the *I–V*. **c** Two simulated *I–V*'s where the black line was obtained with a current source and the red line was obtained with $R_{LOAD}$ = 5.5 kΩ ± 0.1 kΩ. The $R_{LOAD}$ was chosen differently from the experimental curve for demonstration purposes and accounts for the change in slope of red-dashed line between simulation and experimental curves. The red arrows in **b** and **c** indicate the path taken when measuring *I–V* with a triangular $V_{SOURCE}$ sweep. Arrows facing upwards/downwards in current are for increasing/decreasing $V_{SOURCE}$. Source data are provided as a Source Data file

the circuit undergoes a switching event with the current rapidly evolving along the dashed line and reaching a new state at the intersection of the load line and intrinsic device *I–V*. Such a transition, referred to as threshold switching, is volatile and the device returns to its original state after the termination of the voltage pulse. The time necessary for the transition is determined by the heat capacity of the device and is typically in the 0.1 μs–1 μs range for the structures used here[8]. Clearly, based on the above argument, threshold switching occurs at the point when the load

line is tangent to the device *I–V* and the differential circuit resistance vanishes. This is the case for both upward and downward voltage sweeps, which produce a hysteresis in the *I–V*. For a very low load resistance, the transition would occur close to the knee of the intrinsic *I–V*. Figure 1c shows the *I–V* curves corresponding to that in (b) obtained from a finite element model. The continuous black trace is the simulation with a large load resistor included for comparison. It is apparent the simulation correctly describes the hysteresis associated with threshold switching.

**Scanning Joule expansion microscopy.** Our previous simulations predicted a current constriction when the device is biased into the NDR region and stabilized by a large load resistor[7]. Here, we test this assertion experimentally using scanning joule expansion microscopy (SJEM)[20]. In SJEM, an atomic force microscopy (AFM) cantilever mechanically transduces the thermal expansion of the sample induced by the Joule heating that results from biasing the device periodically (see Methods, Supplementary Fig. 2). It should be noted the SJEM signal is proportional to the sample expansion but in general is not an absolute measurement of the expansion. The difficulty of the experiment lies in a relatively long time (~200 s per image) required to scan the device area with the device under bias. For non-optimized devices, local high temperatures within the constriction can allow for ion motion and eventually lead to the device permanently changing its characteristics. To avoid this, we have adjusted the stoichiometry of the functional layer to fabricate devices that are stable for extended time within the NDR region. Also, we have limited the current and time of the experiment to prevent noticeable permanent changes of the device characteristics.

The results are shown in Fig. 2. Figure 2a shows an AFM topographic map of the device, which has a crossbar geometry. The edges of the horizontal TiN bottom electrode extend beyond the bounds of the image and the 6 μm wide gray vertical stripe in the middle corresponds to the TiN top electrode. The 50 nm TaO$_x$ and 15 nm SiO$_2$ layers were sputtered after patterning the bottom electrode and cover an area larger than the scanned region. Before deposition of the top electrode, the SiO$_2$ was removed by ion etching the 2 μm× 2 μm area in the device center to allow a contact between top TiN electrode and the functional layer. This created a depression visible as the dark gray square in the center of the image. This square defines the active area of the device. The devices used in the experiment were nominally identical to devices used in Fig. 1. Figure 2b shows the quasi-DC *I–V* with seven points marked i–vii denoting the points at which SJEM maps were obtained. The maps show a circular feature in the device center with increasing amplitude as both the current and the dissipated power in the device increase. Figure 3a shows the line profiles of the SJEM signal (continuous lines) that allow for a quantitative assessment of the changes. It is apparent that in addition to increasing amplitude, the full width at half maximum (FWHM) of the SJEM signal is decreasing.

The expansion of the device was simulated as part of the finite element model using the following equation:

$$\Delta L(x,y) = \sum_i \frac{1+v_i}{1-v_i} \int_{z_i^0}^{z_i^t} \alpha_i(T(x,y,z) - T_{amb})dz \qquad (2)$$

with *i* numbering all layers in the device structure, $\alpha_i$ corresponding to the coefficient of linear expansion of layer *i*, $v_i$ the Poisson's ratio, *T(x, y, z)* the temperature at point *(x, y, z)*, $T_{amb}$ ambient temperature, and $z_i^0$ and $z_i^t$ refer to the *z*-coordinate of top and bottom of each layer, respectively. We assumed freely expanding boundaries and a fixed temperature of 300 K at the

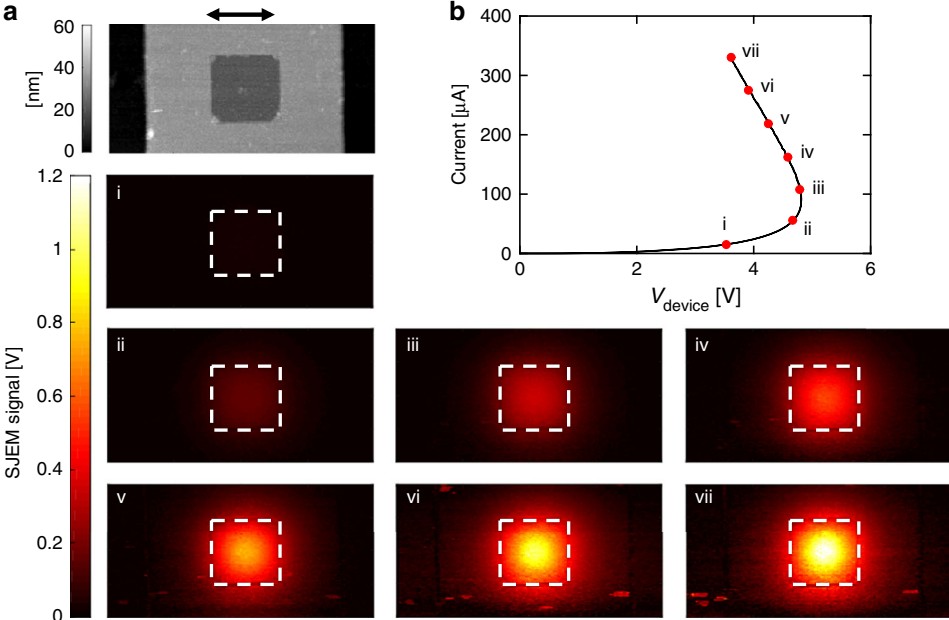

**Fig. 2** SJEM maps measured along the experimental *I–V* curve. **a** AFM topographic map of the TaO$_x$ M/O/M device with the dark square indicating the active area of the device and **b** Quasi-DC *I–V* measured with a load resistance of 95 kΩ ± 1 kΩ. Images marked i–vii present SJEM maps qualitatively highlighting the device thermal expansion at corresponding i–vii points on the *I–V*. Common scale bar above (**a**) is 2 μm. Source data are provided as a Source Data file

bottom of the large slab ($200 \times 100$ μm) used in the simulation. The simulation used the same frequency and power dissipation of the SJEM measurement. The simulated expansion line profiles are shown in Fig. 3a as dashed lines. The SJEM signal is proportional to the expansion and the two were correlated by assuming that the SJEM signal in the center of the device at point vii on the *I–V* corresponds to the value of simulated expansion at this point. This gave the calibration factor for SJEM signal of 1 V nm$^{-1}$ that was used to scale all other SJEM values. The good agreement between simulation and experiment validates the spatial dependencies of current density, temperature, and thermal expansion generated by the finite-element model.

The most important aspect of these distributions is the evolution of all FWHMs. Figure 3b shows the FWHMs of the experimental SJEM, simulated thermal expansion, temperature, and current density profiles as a function of dissipated power. All show gradual narrowing of the distribution as the device is biased deeper into the NDR region, which corresponds to the formation of the current constriction in the S-NDR type device. The simulated FWHM values for current density were 2.26 μm at point i and 0.84 μm at point vii, corresponding to FWHM decreasing by a factor of 2.7 (Supplementary Fig. 3). Current density shows a more pronounced constriction than expansion or temperature because of the exponential dependence of the electrical conductivity on temperature. The ratio of maximum current density at point vii to the value at the periphery of the device was 25. Clearly, the current density within NDR is far from uniform and while it can be described as decomposing into high and low current density domains neither one of those has uniform current or temperature. The constriction occurs gradually with the increase of dissipated power and the transition between high and low current domains is also gradual. This is contrary to the interpretation presented by Kumar and Williams who assumed uniform current density within the domains in construction of *I–V*'s and calculation of internal energies[10].

It should be pointed out that the discussion of current constriction, i.e., decreasing FWHM of current density applies to

the middle part of the S-curve not far from the "knee" of the characteristics. The upper part of the S with positive $\partial I/\partial V$ corresponds in the VO$_2$ devices to a broadening high current density domain, which eventually fills up the entire device[21]. Such point cannot be reached in TaO$_x$-based devices and have not been simulated.

**Multivalued *I–V* characteristics**. The S-NDR shape is not the only type of *I–V* characteristic that can be observed in TaO$_x$-based devices. A nontrivially different *I–V*, shown in Fig. 4a, includes upward and downward sweep of the source voltage for a device with lateral size of 10 μm × 10 μm. The device measured here was fabricated on the same chip as devices shown in Figs. 1 and 2 with all layers having the same thickness and composition. The rise and fall times of the voltage sweep were 2 ms each. The sweep rate was selected to limit the time at temperature and prevent permanent changes taking place at higher current values. At low voltages, the trace is similar to the typical S-NDR; at 3.8 V it 'snaps' along the load line to the high conductivity branch of *I–V*, and shows hysteresis on the downward sweep. Distinct from Fig. 1b, there are two solutions for the voltage when the current is in the 900–1500 μA range, i.e., for any given value of current in this range, there are two possible states of the device with the same current but different device voltages (and presumably different current distributions). The bias history determines which of the possible states the device is in. At low voltages, the downward sweep retraces the upward one indicating that all changes are volatile and the device returns to its original state after the end of the sweep.

The second significant difference between *I–V*'s in Figs. 4a and 1b is that the load line at the snap in the 10 μm × 10 μm device is not tangent to the *I–V* at the point of 'snap', and the sum of the device and the load resistances was positive. This indicates that the transition responsible for the switching is of a different nature than the vanishing total circuit resistance. The load line in Fig. 4a is formed by multiple points collected in the experiment every

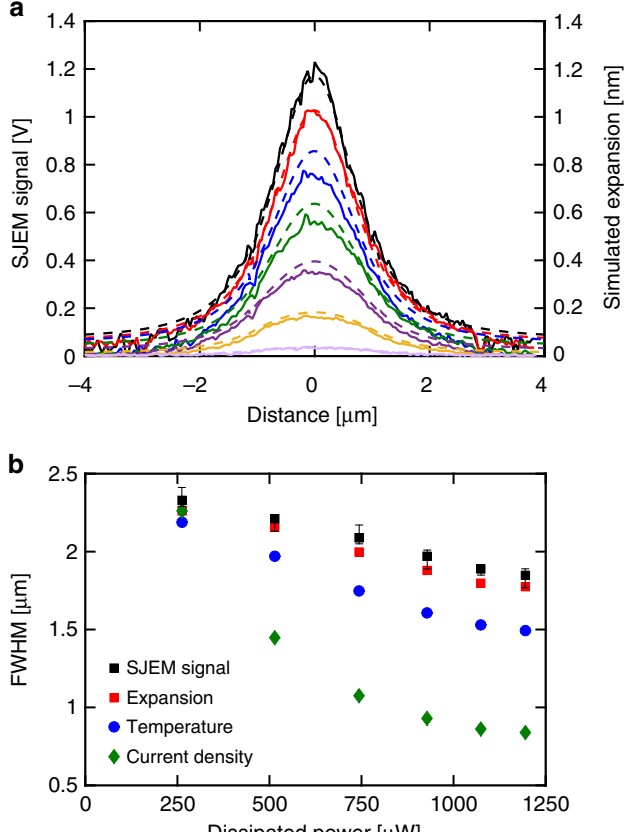

**Fig. 3** Comparison of experimental and simulated current constriction. **a** Line scans obtained from experimental SJEM maps (continuous lines) and simulated (dashed lines) thermal expansion of the TaO$_x$ device within the NDR region. **b** Experimental (black) and simulated values of full width at half maximum of thermal expansion (red square), temperature (blue circle), and current density (green diamond). The error bars in the experimental data represent a single standard deviation in the determination of the FWHM due to the noise present in the SJEM signal. Source data are provided as a Source Data file

referred to as the uniform and constricted current flow branches, respectively. The maximum calculated temperature increases at points A and B are 80 and 480 K, even though the dissipated power at A is considerably higher than that at B.

Figure 4d shows three quasi-DC I–V traces corresponding to devices with lateral size of 10 μm × 10 μm (black trace), 8 μm × 8 μm (red trace), and 2 μm × 2 μm (blue trace), simulated using the same temperature-dependent conductivity. Only the upward sweep of source voltage is plotted. It is apparent that $V_{TH}$ decreases slowly with increasing device size while the current at the knee increases. This is expected since $\partial I/\partial V$ diverges to infinity when the conductivity increase at the hottest point in the device is about twice that in a device at the same voltage but ambient temperature[22]. It takes considerably more power (current) to heat up a large device than a small one to the same critical temperature. The 2 μm × 2 μm device shows a continuous I–V characteristic that corresponds to the lower part of the S-NDR curve. The devices larger than 7 μm × 7 μm, however, show two I–V branches with abrupt transition to the state with the same current but smaller device voltage. The load line is not tangent to I(V) at the point of transition similarly as is the case in the experiment (Fig. 4a). Apparently, with the increase of the source voltage past the knee of the characteristics, the current starts to constrict approaching the current distribution in the constricted branch. At some point, the distribution in uniform branch becomes unstable and the Joule heating in the center of the device runs away, forming a much smaller high current domain. The stability here depends not only on the thermodynamic characteristics of the two steady states but also on the intermediate states of the current density distribution as argued by Landauer[15].

With the increase of device current, the I–V's shown in Fig. 4d converge with the current density distribution converging as well (not shown). Most of the current is flowing through the constriction, which has a size much smaller than the device diameter and the same size for all devices considered here. This "natural" size of the constriction appears to be dependent upon the current in the circuit (determined in large part by the load resistor) and thermal environment, both of which are independent of the device size.

## Discussion

The above observations also explain the conversion of the multivalued I–V into an S-NDR type with the decreasing device size. It occurs when the device size approaches the size of natural constriction in a large device. The boundaries of the device force the current distribution to be close to the natural constriction for all values of device voltage and the evolution along the S-NDR curve is gradual. In a device much larger than the natural constriction, the uniform and constricted solutions are far apart and can co-exist close to the knee of I–V characteristics. As the current increases and the current starts to constrict, the uniform solution becomes unstable and suddenly collapses to the constricted one.

The experimental and simulation results presented above are in contrast with the interpretations proposed by Kumar and Williams[10]. The multivalued-type I–V's were observed in experiments performed on devices where all material parameters, such as electrical and thermal conductivities and heat capacity, change gradually with temperature or remain constant. Such characteristics were also simulated without invoking rapid changes of conductivities or latent heat of the phase transition. The transitions between the two branches in such systems are the consequence of nonlinear dynamics of the heat and charge flow. In particular, the multivalued I–V's were reproduced with

20 ns. The time of the transition is 440 ns ± 20 ns, in agreement with the value published in previous work[8]. The uncertainty in the transition time is determined by the time resolution of the oscilloscope used in the experiment.

Figure 4b displays simulated quasi-DC I–V characteristics for both upward and downward voltage sweeps for a circular device with diameter of 11.28 μm (area equal to that of the 10 μm × 10 μm square device). Except for the lateral size and the value of the load resistance, all devices and material parameters were the same as the ones used in Fig. 1a, which produced the S-NDR characteristics. The load was assumed infinite to ensure that the I–V was not of the S-NDR-type. We were not able to use higher load resistance in the experiment due to instrument voltage limitations. The simulation reproduced the two main features of the I–V: two solutions for voltage were obtained for current values above the knee of the characteristics and the 'snap' occurred at positive values of circuit resistance.

Insight into the nature of the states in the two branches of the I–V can be gained by plotting the current density profiles in the 10 μm device for points A–D in Fig. 4c. The current distributions at A and D are reasonably uniform across the device while at B and C the current flow is sharply constricted with over two orders of magnitude change of current density between the center and the device periphery. The two distinct branches of the I–V will be

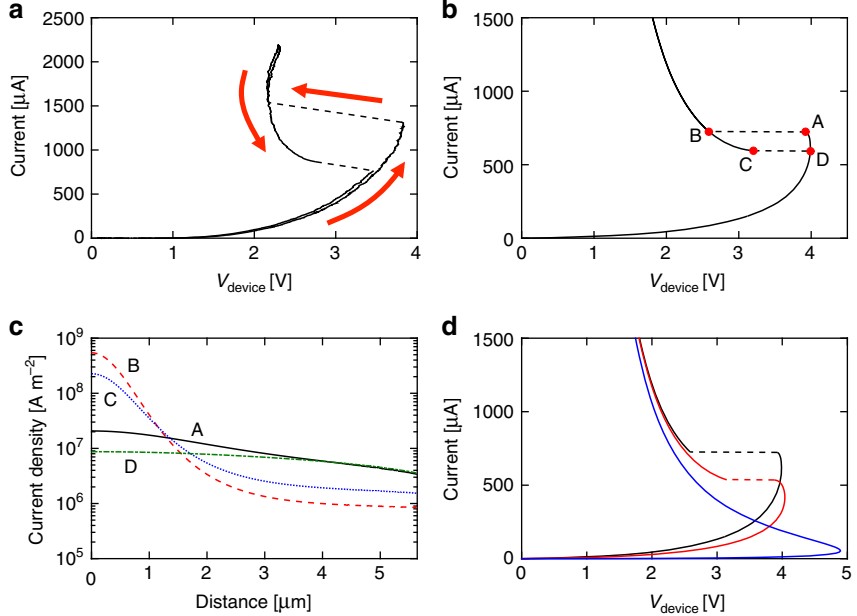

**Fig. 4** Emergence of multivalued *I–V* characteristics in larger devices. **a** Multivalued *I–V* characteristics for 10 µm × 10 µm TaO$_x$ device with $R_{LOAD}$ = 7.5 kΩ ± 0.1 kΩ. Upward arrows indicate increasing source voltage while the downward one indicates decreasing source voltage sweep. **b** Simulated quasi-DC *I–V* with current source for circular device with the same area (11.28 µm diameter) as the device in **a**. **c** Line profiles of current density along the radius of the device for points A–D marked on *I–V* characteristics in **b**. The origin of the horizontal axis corresponds to the center of the circularly symmetric device. **d** Simulations of quasi-DC *I–V*'s with current source for devices with diameter 11.28µm (10 µm × 10 µm—black), 9.02 µm (8 µm × 8 µm—red), and 2.26 µm (2 µm × 2 µm—blue). Source data are provided as a Source Data file

non-uniform current but uniform electric field. We did not have to assume formation of distinct voltage domains. While such domains can, in principle, form in some systems, they are not essential for multivalued *I–V*'s.

As is evident from Fig. 4d, the multivalued-type *I–V* appears in devices 7 µm × 7 µm or larger due to the relatively large natural size of the constriction. While interesting from the fundamental understanding point of view, large devices are of little interest for current semiconductor technology. This prompts the question whether the constriction can occur in small devices and what parameters control the constriction size. A partial answer is provided by simulation of a 200 nm diameter sandwich structure consisting of Au electrodes and a 50 nm thick VO$_2$ layer with conductivity ($\sigma(T)$) adopted from Radu et al.[23]. All other material parameters were assumed to be temperature-independent with unchanged values for all the phases present. In particular, we assumed constant thermal conductivity. While one could expect an increase of conductivity due to changing density of free electrons, recent experimental data indicate that the change is negligible[24]. We also neglect the existence of the hysteresis in the dependence of conductivity on the temperature and the latent heat of phase transition. These effects, if included, would complicate the analysis but would not affect the conclusions. In other words, we focus exclusively on the effects of the electrical conductivity change during IMT.

As seen from Fig. 5, the simulated *I–V* characteristic of the VO$_2$ device is of the multivalued type showing two separate branches above the threshold voltage. Similarly, as for TaO$_x$-based devices, the low conductance branch has a much more uniform current flow than the high conductance branch. The diameters of the constriction at half the maximum of the current density for points B and C are 13 and 7 nm, respectively (Supplementary Fig. 4). The much smaller constriction size, which is comparable to the device size at the 10 nm technology node, is due to the very steep increase of conductivity with temperature and to the high value of conductivity above the IMT.

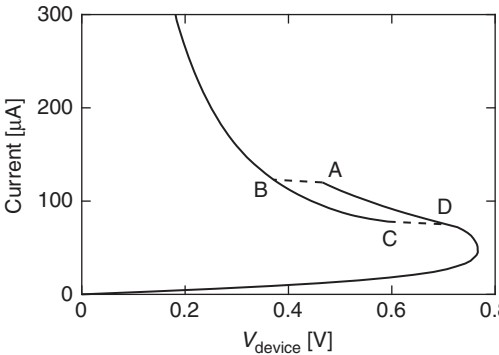

**Fig. 5** Simulated quasi-DC *I–V* of 200 nm diameter VO$_2$ device with current source. Source data are provided as a Source Data file

In summary, we report experimental observation of spontaneous current constriction (electronic decomposition) in TaO$_x$-based devices exhibiting S-NDR. The steady state current density distribution narrows gradually as the device is biased deeper into the NDR region in quantitative agreement with an electro-thermal model. In addition, TaO$_x$ devices with size larger than the "natural" size of the high current density domain exhibit a multivalued *I–V* characteristic with more than one possible solution for voltage and current density for the range of current and voltage values. The proposed electro-thermal model of threshold switching devices consistently predicts phenomena that, so far, have been thought to be of distinctly different origin.

## Methods

**Device fabrication**. Devices were fabricated in a metal/insulator/metal structure with a TiN (40 nm)/TaO$_x$ (50 nm)/TiN (20 nm) stack. The 2 µm × 2 µm active device region was created by etching through a 15 nm SiO$_2$ layer in a larger 6 µm × 6 µm overlapping region. All materials were deposited by sputtering. The TaO$_x$ layer was reactively sputtered with 3 × 10$^{-8}$ m$^3$ s$^{-1}$ (2 sccm) O$_2$ flow into the

chamber during Ta deposition at 4 Pa. The bottom electrode was formed by patterning a blanket TiN layer. All other layers were formed using a lift-off process. An on-chip serpentine resistor was patterned near the device region bottom electrode.

**SJEM**. SJEM experiments were conducted in contact-mode using commercially available, uncoated Si cantilevers with a nominal resonance frequency in air of 13 kHz ± 4 kHz and a nominal spring constant in the range of 0.07–0.4 N m$^{-1}$. The cantilever contact resonance frequency was in the range of 54 kHz ± 4 kHz depending on the local tip-sample contact. SJEM maps were obtained with 0.25 Hz scan rate and 40 nm/pixel resolution while biasing the devices with a 95 kHz square-wave voltage (50% duty cycle) of amplitude $V_s$ (Supplementary Fig. 2) and grounding the device top electrode. The circuit included the voltage source, on-chip series resistor (95 kΩ ± 1 kΩ), and device connected in series. The applied voltage induces a periodic expansion of the entire device structure due to Joule heating which pushes the cantilever into oscillation with amplitude proportional to the device expansion (the AFM cantilever motion mechanically amplifies the thermal expansion of the device). The AFM detector measures the AFM cantilever oscillation at the applied voltage frequency using a lock-in amplifier. The voltage modulation frequency was chosen well above the cantilever contact resonance frequency to minimize the effect of the mechanical and electrostatic coupling between the tip and the device top electrode on the SJEM signal intensity. The 95 kHz modulation frequency corresponds to 5.2 µs of heating which exceeded the device thermal time constant (~2 µs), ensuring the steady-state temperature distribution in the device was reached during each cycle.

**Electrothermal simulation of device I–V and expansion**. In this study, we simulated threshold switching devices using a commercial finite element simulation software package (COMSOL Multiphysics). Certain commercial equipment, instruments, or materials are identified in this paper to foster understanding. Such identification does not imply recommendation or endorsement by the National Institute of Standards and Technology, nor does it imply that the materials or equipment identified are necessarily the best available for the purpose. The simulation solved the 3D equations for the heat and charge flow with an added restriction of the load resistor in the circuit. The thermal expansion of the device structure was calculated with a constant temperature boundary on the bottom of the sample and all boundaries free to expand. The simulation used the same frequency and amplitude of the applied bias during SJEM measurement. We assumed 2-D axisymmetric device structure connected to the voltage source and load resistor. The boundary conditions are shown (Supplementary Fig. 5) and the input material parameters are listed (Supplementary Table 1).
The electrical conductivity of TaO$_x$ was described by Poole-Frenkel formula:

$$\sigma_{PF}(E, T) = \frac{\sigma_0(T)}{E}\left(\frac{k_B T}{\beta}\right)^2 \left\{ 1 + \left(\frac{\beta\sqrt{E}}{k_B T} - 1\right)\exp\left(\frac{\beta\sqrt{E}}{k_B T}\right)\right\} + \frac{\sigma_0(T)}{2} \quad (3a)$$

where

$$\sigma_0(T) = q\mu N_c\left(\frac{N_d}{N_t}\right)^2 \exp\left(-\frac{E_d + E_t}{2k_B T}\right) \quad (3b)$$

$$\beta = \left(\frac{q^3}{\pi\varepsilon_0\varepsilon_i}\right)^{\frac{1}{2}} \quad (3c)$$

and constants $N_d$, $E_d$ and $N_t$, $E_t$ correspond to the densities and ionization energies of donors and traps, respectively, $N_c$ is the effective density of states in the conduction band, $\sigma_0(T)$ is the low field conductivity, $\mu$ is the electron mobility, $k_B$ is the Boltzmann constant, $q$ is the elementary charge, $\varepsilon_0$ is the permittivity of free space, $\varepsilon_i$ is the relative dielectric constant of the material, $E$ is the electric field, and $T$ is the stage temperature. The values used in the simulation were $\sigma_0(T) = 800$ S m$^{-1}$ and $(E_d + E_t)/2 = 0.3$ eV, obtained by fitting to known values[7].

## Data availability

The source data are provided as a source data file (Figs. 1–5, Supplementary Figs. 4, 5). All data are also available by request to the corresponding author.

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

## Acknowledgements

We thank the CMU Nanofab staff for their support in device development. This work was supported in part by NSF Grant DMR 1409068 and the Data Storage Systems Center at Carnegie Mellon University. G.R. acknowledges support under the Cooperative Research Agreement between the University of Maryland and the National Institute of Standards and Technology Center for Nanoscale Science and Technology, Award 70NANB14H209, through the University of Maryland.

## Author contributions

J.M.G., D.L., J.A.B., and M.S. conceived the experiment. J.M.G. and D.L. designed, fabricated and electrically tested all devices and performed finite element simulations. J.M.G., D.L., B.D.H., G.R., G.P., J.J.M., and A.C. designed and conducted the SJEM measurements. All contributors discussed the results and provided input to the manuscript.

## Additional information

**Competing interests:** The authors declare no competing interests.

