## [Peer Review File · Nature Communications]

Reviewers' comments:

Reviewer #2 (Remarks to the Author):

1. In Fig. 1(a), could the authors explain the "super linearity at higher voltages"? Also, In Fig. 1, it is mentioned that smaller RLOAD induces hysteresis behaviour of the I-V. Then, could you comment on how to choose RLOAD to induce optimum hysteresis associated with threshold switching.
2. Could the authors explain why there is a dark square contrast at the centre of the height map of Fig. 2 (b)?
3. In Fig. 3(a), how can one correlate the experimental SJEM signals in voltage to that of the simulated expansion in nm?
4. In line number 98 and 99, it has been highlighted that "At low bias the I-V is linear, becoming super linear at higher voltages due to Joule heating and the reduction of effective activation energy of traps in the Poole-Frenkel model of conductivity". In this statement, when you mention super linear at high voltage, is current flow constriction due to oxygen vacancies? Do you see any role of metal migration from the electrodes if the joule heating is too high?
5. Starting line number 161, it is mentioned that "we have carefully adjusted the TaOx stoichiometry by controlling the oxygen flow during deposition of the functional layer. This, in turn, controls the activation energy of the conductivity and the constriction's size and temperature".
6. Why a triangular voltage pulse instead of square pulse was used for the experimental data shown Fig. 4 (a)? Also, the snap seemed to occur at 3.8V instead of 3.1V as mentioned in the manuscript (page XIII, line 215).
7. Can the authors define clearly the distance shown in Fig. 4(c). Perhaps Fig. S2 can be used as a reference.
8. Line 323, can the author explain the type of metallic phase? Is this the metal filament formed within the thick TaOx?
9. The authors should mention the name of the commercial finite element simulator (Supplementary Materials, line 8).
10. Fig. S2 is unclear. What do the dimensions represent?
11. Fig. S5(c), the results should be plotted with an expanded view to see clearly of the results of A.
12. Is same TaOx stoichiometry maintained for devices with different sizes? If not, how it is related to device dimensions and affect the working of the device?
13. Starting from line number 293, it is stated that "The multivalued-type I-V's were observed in devices where all material parameters change gradually with temperature, if at all". Could you explain specifically what material parameters change gradually with temperature?

Reviewer #3 (Remarks to the Author):

This is an interesting article revealing important new insights in the physics of threshold switching,

so I would strongly recommend it for publication in Nature Communications.

They are only a number of (relatively minor) comments that I think should be addressed in order to clarify better some aspects of this paper:

1)Page I (Abstract):

"the current spontaneously and gradually constricts" What does this mean? Seems to be saying two opposite things.

2)Page VI:

"can be retraced many times" What's many times? Please quantify (a number like 10^x times) ?

3)Page VI:

"The steady state solution for the I(V) faithfully reproduces the shape of the I-V."
A bit of an overstatement. The agreement is approximate.

4)

Page VII:

Fig 1.(a). Why is the simulation so inaccurate? probably because "The simulations did not use any adjustable parameters." so in this context it is actually pretty accurate. This should be better pointed out in the paper.

5)Page VII: Fig.1

Why are the black curves simulations in (b) and (c)? They supposedly measured the NDR curve in (a) so why not compare to that?

6)Page X

Fig.2: One should be very careful in using the rainbow color map.

See: D. Borland and R. M. Taylor II, "Rainbow Color Map (Still) Considered Harmful," in IEEE Computer Graphics and Applications, vol. 27, no. 2, pp. 14-17, March-April 2007.

This distorts the contrast and makes a current constriction appear (making the transition look much more abruptly than it actually is).

A perceptually uniform color map could change the interpretation completely.

If possible, change this map.

7)Page XI:

" Narrowing of distributions as the device goes "deeper" into the NDR Region" contradicts some of their earlier claims of the authors [1] that the current constriction eventually "fills up" the device and we back to PDR

[1] Li et al "Scaling behavior of oxide-based electrothermal threshold switching devices" Nanoscale 2017

So, which one is it?

8)Page XII:

Fig.3 (b) needs a legend for the colors. I don't want to have to read the caption to see what the plot is.

9)Page XVI:

Not obvious why there should be a "natural" domain size. What controls this?

10)Page XVII:

In the discussion with the results of Kumar and Williams I don't see why both can't be true (although the one reported by Kumar may be a measurement artifact, the effect could happen in principle)

11)Page XVIII

They can also observe "multi-valued" effect in simulation for smaller devices "The much smaller constriction size ... is due to the very steep increase of conductivity with temperature and to the high value of the conductivity in metallic phase, among other factors."

More general, it would be interesting to investigate how the feedback of conduction mechanism affects the heat profile ? And how the strength of this feedback relates to the size of the constriction so we can know from which size of devices they may occur?

12)Page XX

Apparently they did not use the on-chip resistor of 95kohm? Where is the explanation of the electrical setup?

13) what is the thickness of the films? 50nm only revealed in the experimental section on page XX, should be sooner

We cite reviewer's comments verbatim in black font in italics. Our responses are in blue. The cited fragments of the manuscript are in quotation marks with the original text in black and modifications highlighted in red.

Reviewers' comments:

Reviewer #2 (Remarks to the Author):

1. In Fig. 1(a), could the authors explain the “super linearity at higher voltages”?

The super linear dependence of current in oxide-based devices is common and extensive comment is unnecessary. We have added a short statement following reviewer's suggestion on page V:

“Amorphous oxide-based devices typically display exponential dependence of current on applied voltage and thermally activated dependence on temperature. This behavior is frequently fitted with Poole-Frenkel model of conductivity^{17,18}. This corresponds to linear I - V at low bias, becoming visibly super linear at voltages exceeding 2.5 V due to Joule heating and the field-induced reduction of effective activation energy of traps^{7,19}. There is an obvious positive feedback between the current, conductance, and temperature of the device. With further increase in device current...”

Also, In Fig. 1, it is mentioned that smaller R_{LOAD} induces hysteresis behavior of the I - V . Then, could you comment on how to choose R_{LOAD} to induce optimum hysteresis associated with threshold switching.

The dependence of hysteresis on R_{LOAD} is quite apparent if the intrinsic I - V characteristics of the selector (threshold switch) are known. Also, the load is typically in the form of the resistive switching device connected in series with the selector leaving no choice for the R_{LOAD} . For these reasons, we feel an analysis of this sort is beyond the scope of this paper.

2. Could the authors explain why there is a dark square contrast at the centre of the height map of Fig. 2 (b)?

The following text was added to the description of the figure on page X:

“The results are shown in Figure 2. Figure 2(b) shows an Atomic Force Microscopy topographic map of the device, which has a crossbar geometry. The edges of the horizontal TiN bottom electrode extend beyond the bounds of the image and the 6 μm wide grey vertical stripe in the middle corresponds to the TiN top electrode. The 50 nm TaO_x and 15 nm SiO_2 layers were sputtered after patterning the bottom electrode and cover an area larger than the scanned region. Before deposition of the top electrode, the SiO_2 was removed by ion etching the $2 \times 2 \mu\text{m}$ area in the device center to allow a contact between top TiN electrode and the functional layer. This created a depression visible as the dark grey square in the center of the image. This square defines the active area of the

device. The devices used in the experiment were nominally identical to devices..."

3. In Fig. 3(a), how can one correlate the experimental SJEM signals in voltage to that of the simulated expansion in nm?

We have modified the description of Figure 3(a) on page XI by providing more detail of the simulation and the calibration procedure. The modified text is highlighted in the manuscript and is included below:

"The expansion of the device was simulated as part of the finite element model using the following equation:

$$\Delta L(x, y) = \sum_i \frac{1+\nu_i}{1-\nu_i} \int_{z_i^0}^{z_i^t} \alpha_i (T(x, y, z) - T_{amb}) dz \quad (4)$$

with i numbering all layers in the device structure, α_i corresponding to the coefficient of linear expansion of layer i , ν_i the Poisson's ratio, $T(x,y,z)$ the temperature at point (x,y,z) , T_{amb} ambient temperature, and z_i^0 and z_i^t refer to the z-coordinate of top and bottom of each layer, respectively. We assumed freely expanding boundaries and a fixed temperature of 300 K at the bottom of the large slab ($200 \times 100 \mu\text{m}$) used in the simulation. The simulation used the same frequency and power dissipation of the SJEM measurement. The simulated expansion line profiles are shown in Figure 3(a) as dashed lines. The SJEM signal is proportional to the expansion and the two were correlated by assuming that the SJEM signal in the center of the device at point vii on the I - V corresponds to the value of simulated expansion at this point. This gave the calibration factor for SJEM signal of 1 V/nm that was used to scale all other SJEM values. The good agreement..."

4. In line number 98 and 99, it has been highlighted that "At low bias the I - V is linear, becoming super linear at higher voltages due to Joule heating and the reduction of effective activation energy of traps in the Poole-Frenkel model of conductivity". In this statement, when you mention super linear at high voltage, is current flow constriction due to oxygen vacancies? Do you see any role of metal migration from the electrodes if the joule heating is too high?

The current constriction observed in this work is a result of the positive feedback loop between thermally activated Poole-Frenkel conductivity and Joule heating. The current constriction is not a result of any permanent changes within the device such as redistribution of either oxygen or tantalum. We have clarified this with the following additions to the text:

On page V:

"Amorphous oxide-based devices typically display exponential dependence of current on applied voltage and thermally activated dependence on temperature. This behavior is frequently fitted with Poole-Frenkel model of conductivity^{17,18}. This corresponds to linear I - V at low bias, becoming visibly super linear at

voltages exceeding 2.5 V due to Joule heating and the field-induced reduction of effective activation energy of traps in Poole-Frenkel model^{7,19}. There is an obvious positive feedback between the current and conductance as well as temperature of the device. With further increase in device current..."

and on page IX:

"...The difficulty of the experiment lies in a relatively long time (≈ 200 s per image) required to scan the device area with the device under bias. For non-optimized devices, local high temperatures within the constriction can allow for ion motion and eventually lead to the device permanently changing its characteristics. To avoid this, we have adjusted the stoichiometry of the functional layer to fabricate devices that are stable for extended time within the NDR region. Also, we have limited the current and time of the experiment to prevent noticeable permanent changes of the device characteristics."

5. Starting line number 161, it is mentioned that "we have carefully adjusted the TaO_x stoichiometry by controlling the oxygen flow during deposition of the functional layer. This, in turn, controls the activation energy of the conductivity and the constriction's size and temperature".

The message we were trying to convey here is that the device needed to withstand high current in the NDR region without any permanent changes long enough to allow for completion of the SJEM scan. In order to do so, we had to specifically design the functional layer of the device by adjusting its stoichiometry. We have clarified this point on page IX:

"...The difficulty of the experiment lies in a relatively long time (≈ 200 s per image) required to scan the device area with the device under bias. For non-optimized devices, local high temperatures within the constriction can allow for ion motion and eventually lead to the device permanently changing its characteristics. To avoid this, we have adjusted the stoichiometry of the functional layer to fabricate devices that are stable for extended time within the NDR region. Also, we have limited the current and time of the experiment to prevent noticeable permanent changes of the device characteristics."

6. Why a triangular voltage pulse instead of square pulse was used for the experimental data shown Fig. 4 (a)? Also, the snap seemed to occur at 3.8V instead of 3.1V as mentioned in the manuscript (page XIII, line 215).

The expression "triangular voltage pulse" is somewhat confusing and was replaced in the revised manuscript by "voltage sweep." The entire paragraph was rephrased to avoid misunderstandings on page XIV:

"The S-NDR shape is not the only type of I - V characteristic that can be observed in TaO_x-based devices. A nontrivially different I - V , shown in Figure 4(a), includes upward and downward sweep of the source voltage for a device with lateral size of $10\ \mu\text{m} \times 10\ \mu\text{m}$. The device measured here was fabricated on the same chip as devices shown in Figures 1 and 2 with all layers having the same thickness

and composition. The rise and fall times of the voltage sweep were 2 ms each. The sweep rate was selected to limit the time at temperature and prevent permanent changes taking place at higher current values. At low voltages..."

The reviewer is correct, the voltage is 3.8 V instead of 3.1 V, and has been changed accordingly.

7. *Can the authors define clearly the distance shown in Fig. 4(c). Perhaps Fig. S2 can be used as a reference.*

The caption of Figure 4(c) has been modified to clarify the distance on the x-axis as follows:

"(c) Line profiles of current density along the radius of the device for points A-D marked on *I-V* characteristics in (b). The origin of the horizontal axis corresponds to the center of the circularly symmetric device."

8. *Line 323, can the author explain the type of metallic phase? Is this the metal filament formed within the thick TaOx?*

The metallic phase mentioned here is in reference to the metallic phase of VO₂ above the insulator-metal transition temperature. The main concept we want to convey is the small radius of current constriction in VO₂ is due to the steep dependence of conductivity on temperature. Therefore, to avoid confusion we have replaced the phrase on page XIX:

"...The much smaller constriction size, which is comparable to the device size at the 10 nm technology node, is due to the very steep increase of conductivity with temperature and to the high value of conductivity above the IMT."

9. *The authors should mention the name of the commercial finite element simulator (Supplementary Materials, line 8).*

The name of the finite element software has been listed in the Supplementary material.

10. *Fig. S2 is unclear. What do the dimensions represent?*

A statement specifying the dimensions has been added to the Figure S2 caption:

"The dimensions shown for each layer are indicated as width × height."

11. *Fig. S5(c), the results should be plotted with an expanded view to see clearly of the results of A.*

Figure S5(c) has been modified: the linear scale has been replaced with a logarithmic one to highlight the small changes in line profile A.

12. *Is same TaOx stoichiometry maintained for devices with different sizes? If not, how it is related to device dimensions and affect the working of the device?*

Devices of different sizes were fabricated on the same chip and had the same deposition conditions and TaO_x stoichiometry. We have added a statement on page XIV to clarify:

"...A nontrivially different I - V , shown in Figure 4(a), includes upward and downward sweep of the source voltage for a device with lateral size of 10 μm \times 10 μm . The device measured here was fabricated on the same chip as devices shown in Figures 1 and 2 with all layers having the same thickness and composition. The rise and fall times of the voltage sweep were 2 ms each. The sweep rate was selected to limit the time at temperature and prevent permanent changes taking place at higher current values. At low voltages..."

13. Starting from line number 293, it is stated that "The multivalued-type I - V 's were observed in devices where all material parameters change gradually with temperature, if at all". Could you explain specifically what material parameters change gradually with temperature?

A statement has been added on page XVIII to clarify the material properties in question:

"The experimental and simulation results presented above are in sharp contrast with the interpretations advanced by Kumar and Williams¹⁰. The multivalued-type I - V 's were observed in devices where all material parameters, such as electrical and thermal conductivities and heat capacity, change gradually with temperature or remain constant. Such characteristics were also simulated without invoking rapid changes of conductivities or latent heat of the phase transition..."

Reviewer #3 (Remarks to the Author):

This is an interesting article revealing important new insights in the physics of threshold switching, so I would strongly recommend it for publication in Nature Communications.

They are only a number of (relatively minor) comments that I think should be addressed in order to clarify better some aspects of this paper:

1) Page I (Abstract): "the current spontaneously and gradually constricts" What does this mean? Seems to be saying two opposite things.

The word "spontaneous" means for something to happen without any apparent external cause. The same word is also sometimes used in lieu of "sudden." We use it in the manuscript as "without apparent cause." We have modified the abstract to clarify this point on page I:

"...It has been proposed that such non-linear characteristics are associated with a spontaneous current flow constriction i.e. formation of high current density domains that are volatile and dissolve with the termination of bias. Spontaneous

is used to indicate a phenomenon that occurs without an apparent external stimulus such as a defect or inhomogeneity in the device. The size and density of such domains and the mechanism underlying their formation is currently a subject of debate..."

2) Page VI: "can be retraced many times" What's many times? Please quantify (a number like 10^X times) ?

We have included the following changes on page VI to indicate the cycling ability of the devices:

"...Due to the large series load resistor, the total differential circuit resistance was always positive, the current was well defined, and changed gradually with the increase of V_{SOURCE} . The I - V curve displays no noticeable changes up to 10^9 cycles if the current is limited to less than $350 \mu A$ and shows symmetry with respect to the bias polarity."

3) Page VI: "The steady state solution for the $I(V)$ faithfully reproduces the shape of the I - V ." A bit of an overstatement. The agreement is approximate.

The offending sentences have been modified on page VI:

"...The steady state solution for the $I(V_{DEVICE})$ reasonably well reproduces the shape of the I - V , especially considering that the simulation did not include any adjustable parameters."

4) Page VII: Fig 1.(a). Why is the simulation so inaccurate? probably because "The simulations did not use any adjustable parameters." so in this context it is actually pretty accurate. This should be better pointed out in the paper.

Addressed in response to the comment above.

5) Page VII: Fig.1 Why are the black curves simulations in (b) and (c)? They supposedly measured the NDR curve in (a) so why not compare to that?

Reviewer is incorrect in this comment. Figure 1(b) shows two experimental I - V 's with different load resistances while Figure 1(c) shows two simulated curves with different load resistance. We have changed the Figure 1 caption to clarify this point:

"(b) Magnified view of two experimental I - V 's with different values of load resistance. The black line is the I - V from (a) and the red line is the I - V obtained with $R_{LOAD} = 3.9 \text{ k}\Omega \pm 0.1 \text{ k}\Omega$. The smaller R_{LOAD} induces threshold switching (red dashed lines) with hysteretic behavior of the I - V . (c) Two simulated I - V 's where the black line was obtained with a current source and the red line was obtained with $R_{LOAD} = 5.5 \text{ k}\Omega \pm 0.1 \text{ k}\Omega$."

6) Page X Fig.2: One should be very careful in using the rainbow color map. See: D. Borland and R. M. Taylor II, "Rainbow Color Map (Still) Considered

Harmful," in IEEE Computer Graphics and Applications, vol. 27, no. 2, pp. 14-17, March-April 2007.

This distorts the contrast and makes a current constriction appear (making the transition look much more abruptly than it actually is). A perceptually uniform color map could change the interpretation completely. If possible, change this map.

The color maps in Figure 2 have been changed according to reviewer suggestion.

7) Page XI: "Narrowing of distributions as the device goes "deeper" into the NDR Region" contradicts some of their earlier claims of the authors [1] that the current constriction eventually "fills up" the device and we go back to PDR [1] Li et al "Scaling behavior of oxide-based electrothermal threshold switching devices" Nanoscale 2017. So, which one is it?

The reviewer is correct that the paper by Li et al. states that at high currents the current density domain starts to broaden and eventually fills the entire device. This is what gives rise to the upper part of S-type curve, and more specifically, the 2nd turnover where the differential resistance becomes positive again. Since TaO_x devices are limited to low current densities, we could not reach this point in the experiment and refrained from extending the model beyond physical limits of the device. We have added the following paragraph to illustrate this point on page XII:

"It should be pointed out that the discussion of current constriction i.e. decreasing FWHM of current density applies to the middle part of the S-curve not far from the "knee" of the characteristics. The upper part of the S with positive $\partial I/\partial V$ corresponds in the VO₂ devices to a broadening high current density domain, which eventually fills up the entire device²¹. Such point cannot be reached in TaO_x-based devices and have not been simulated."

8) Page XII: Fig.3 (b) needs a legend for the colors. I don't want to have to read the caption to see what the plot is.

A legend was added to Figure 3(b).

9) Page XVI: Not obvious why there should be a "natural" domain size. What controls this?

The "natural" domain size is controlled by the conductivity dependence on temperature. This is evident by the *I-V* characteristics converging at higher current values in Figure 4(d), which is indicative of the similarity in current distributions in the devices regardless of the device lateral size. The conversion from a multivalued to a S-NDR type *I-V* is based the thermal environment and the stability with which the device can sustain a broad distribution of elevated current density. We have addressed this with the following changes on page XVII and XVII:

"With the increase of device current, the I - V 's shown in Figure 4(d) converge with the current density distribution converging as well (not shown). Most of the current is flowing through the constriction, which has a size much smaller than the device diameter and the same size for all devices considered here. This "natural" size of the constriction appears to be dependent upon the current in the circuit (determined in large part by the load resistor) and thermal environment, both of which are independent of the device size.

The above observations also explain the conversion of the multivalued I - V into an S-NDR type with the decreasing device size. It occurs when the device size approaches the size of natural constriction in a large device. The boundaries of the device force the current distribution to be close to the natural constriction for all values of device voltage and the evolution along the S-NDR curve is gradual. In a device much larger than the natural constriction, the uniform and constricted solutions are far apart and can co-exist close to the knee of I - V characteristics. As the current increases and the current starts to constrict, the uniform solution becomes unstable and suddenly collapses to the constricted one.

10) Page XVII: *In the discussion with the results of Kumar and Williams I don't see why both can't be true (although the one reported by Kumar may be a measurement artifact, the effect could happen in principle)*

We agree with the reviewer that the high voltage domains suggested by Kumar and Williams can, in principle, form in some devices. It was not our intention to claim that this is impossible. We were making a point that they are not necessary. We have added a following sentence to the discussion on page XVIII:

"...In particular, the multivalued I - V 's were reproduced with non-uniform current but uniform electric field. We did not have to assume formation of distinct voltage domains. While such domains can, in principle, form in some systems, they are not essential for multivalued I - V 's.

11) Page XVIII: *They can also observe "multi-valued" effect in simulation for smaller devices "The much smaller constriction size ... is due to the very steep increase of conductivity with temperature and to the high value of the conductivity in metallic phase, among other factors."*

More general, it would be interesting to investigate how the feedback of conduction mechanism affects the heat profile? And how the strength of this feedback relates to the size of the constriction so we can know from which size of devices they may occur?

The size of the constriction is dependent upon both the dependence of electrical conductivity on temperature and the thermal environment of the device. The dependence, however, is quite complex and an adequate discussion of this could not be included in this manuscript due to page limitations. It is an interesting subject that we hope to address in future work.

12) Page XX: Apparently they did not use the on-chip resistor of 95kohm? Where is the explanation of the electrical setup?

The SJEM electrical set-up was the same as for devices shown in Figure 1 but with a different value of load resistance. To clarify this, a statement has been added to Figure 2 caption on page X:

“Figure 2. (a) Quasi-DC *I-V* measured with a load resistance of $95\text{ k}\Omega \pm 1\text{ k}\Omega$ and (b) AFM topographic map of the TaO_x M/O/M device with the dark square indicating the active area of the device. Images marked i-vii present SJEM maps qualitatively highlighting the device thermal expansion at corresponding i-vii points on the *I-V*.”

and in the SJEM experimental methods on page XXI.

“...SJEM maps were obtained with 0.25 Hz scan rate and 40 nm/pixel resolution while biasing the devices with a 95 kHz square-wave voltage (50% duty cycle) of amplitude V_s (see Figure S3) and grounding the device top electrode. The circuit included the voltage source, on-chip series resistor ($95\text{ k}\Omega \pm 1\text{ k}\Omega$), and device connected in series. The applied voltage induces a periodic expansion of the entire device structure due to Joule heating which pushes the cantilever into oscillation with amplitude proportional to the device expansion...”

13) what is the thickness of the films? 50nm only revealed in the experimental section on page XX, should be sooner

The suggested information was added on page V:

“Figure 1(a) shows the quasi-DC *I-V* characteristics of a TiN/TaO_x/TiN via-type device with the lateral size of $2\text{ }\mu\text{m} \times 2\text{ }\mu\text{m}$ and a functional layer thickness of 50 nm measured in a circuit that included a $107\text{ k}\Omega \pm 1\text{ k}\Omega$ load resistor. Throughout the manuscript, the resistance uncertainty represents a single standard deviation in fitting the load resistor *I-V* slope...”

REVIEWERS' COMMENTS:

Reviewer #2 (Remarks to the Author):

I have read the revised copy and responses to the reviewers' comments carefully.

The authors have addressed all my comments appropriately and hence I would recommend the manuscript is ready for publication.

Reviewer #3 (Remarks to the Author):

The authors have responded in detail to my previous review comments and modified the text where appropriate.

At this point, I do not have further comments and support publication.